# BCS-BEC crossover driven by small Fermi pockets of a high-$T_c$ cuprate superconductor

Junhyeok Jeong [1], Yamato Enomoto[2], Yoshimitsu Kohama [1], Tomotaka Nakayama[2], Kotaro Ando[2], Kifu Kurokawa[1], Soonsang Huh [1], Zhuo Yang [1], Toshihiro Nomura[3], Matthew D. Watson [4], Timur K. Kim [4], Cephise Cacho [4], Chun Lin [5], Makoto Hashimoto [5], Donghui Lu [5], Shiro Sakai[6], Takami Tohyama [7], Kazuyasu Tokiwa [2] ✉ & Takeshi Kondo [1,8] ✉

Fermi arcs observed in underdoped cuprates have sparked debate over whether they represent segments of a large Fermi surface or small Fermi pockets. This ambiguity has long hindered their classification as either the conventional Bardeen-Cooper-Schrieffer (BCS) regime or the strongly coupled Bose-Einstein condensation (BEC) crossover limit. Here, using angle-resolved photoemission spectroscopy and quantum oscillations, we demonstrate the coexistence of a small Fermi pocket and a large superconducting gap in the clean inner $CuO_2$ layers of the four-layer cuprate $Ba_2Ca_3Cu_4O_8(F,O)_2$. This coexistence constitutes a hallmark of the BCS-BEC crossover and has remained elusive for decades. Despite the presence of antiferromagnetic (AF) order, the superconducting gap in the small pocket is remarkably large, yielding a gap-to-Fermi energy ratio ($\Delta_{pocket}/\varepsilon_F \sim 0.6$) and a critical-to-Fermi temperature ratio ($T_c/T_F \sim 0.13$) that reach the theoretical upper bound for two-dimensional superconductivity. Unexpectedly, this BCS-BEC crossover emerges not as the carrier density decreases but as it increases, abruptly within a narrow doping range of less than 1%. These results provide a long-sought microscopic foundation for the $d$-wave pairing mechanism in doped AF-Mott insulators.

The distinction between a large Fermi surface and a small Fermi pocket with carrier density $1 + p$ and $p$, respectively, remains one of the most fundamental unsolved issues in high-$T_c$ cuprates[1,2]. The carrier density is traditionally estimated from the inverse Hall coefficient via the Drude formula ($R_H = 1/ne$, where $n$ and $e$ are the carrier density and elementary charge), which assumes an isotropic, closed Fermi surface within the Boltzmann transport framework. This "effective" carrier density offers a simplified view of the electronic phase diagram. However, this picture stands in apparent contradiction to angle-resolved photoemission spectroscopy (ARPES) results, which reveal not a closed Fermi surface but rather an anomalous Fermi arc[3]. Because the Fermi arc may represent a fragment of either a large surface or a small pocket, the carrier density in the arc state cannot be determined as either $1 + p$ or $p$[4,5]. Notably, a large Hall coefficient (indicative of small pockets) has been explained by a theory invoking anisotropic scattering on a large Fermi surface[6]. This theory may also be relevant to recent angle-dependent magnetoresistance results[7] presented as evidence for hole pockets. The longstanding issue, expressed as "a large Fermi surface vs. a small Fermi pocket", is apparently not easy to resolve.

[1]Institute for Solid State Physics, The University of Tokyo, Kashiwa, Japan. [2]Department of Applied Electronics, Tokyo University of Science, Tokyo, Japan. [3]Department of Physics, Faculty of Science, Shizuoka University, Shizuoka, Japan. [4]Diamond Light Source Ltd, Harwell Science and Innovation Campus, Didcot, UK. [5]Stanford Synchrotron Radiation Lightsource, SLAC National Accelerator Laboratory, Menlo Park, CA, USA. [6]Faculty of Science and Technology, Sophia University, Tokyo, Japan. [7]Department of Applied Physics, Tokyo University of Science, Tokyo, Japan. [8]Trans-scale Quantum Science Institute, The University of Tokyo, Bunkyo-ku, Japan. ✉e-mail: tokiwa@rs.tus.ac.jp; kondo1215@issp.u-tokyo.ac.jp

Similarly, the presence of the Fermi arc obscures the Fermi energy ($\varepsilon_F$) value, which is estimated from the corresponding band bottom or top. When the Fermi arc is assumed as a fragment of the large Fermi surface centered at ($\pi$, $\pi$), $\varepsilon_F$ is considered to be a substantially large value of approximately 1 eV[8]. In such a case, the pairing strength $\Delta/\varepsilon_F \sim d_p/\xi$ (where $\Delta$ is the energy gap), which describes the ratio between the distance between pairs $d_p$ and the pair size $\xi$ (or coherence length), is estimated to be very small (around 0.03 for a typical gap size of ~30 meV), placing it within the weak-coupling regime. In contrast, if the Fermi arc is part of a small pocket, $\varepsilon_F$ should be much smaller (at least an order of magnitude lower), resulting in a relatively large $\Delta/\varepsilon_F$, which could lie in the BCS-BEC crossover regime. This stark discrepancy, arising from the interpretation of the Fermi arc, presents a major challenge in modeling the pairing mechanism in cuprates.

Most prior studies have focused on single- and double-layer cuprates due to their structural simplicity and relative ease of crystal growth. However, in these systems, disorder is inevitably introduced into the $CuO_2$ superconducting layers owing to their proximity to the dopant layers, which contain random potentials associated with elemental vacancies[9]; this produces inhomogeneous electronic states, as revealed by scanning tunneling microscopy[9,10]. Such disorder leads to a serious mismatch between idealized theoretical models and real materials, as most theories neglect the effects of disorder. Without resolving this discrepancy, a breakthrough in formulating a microscopic theory of pairing may not be achieved. In particular, elucidating the electronic nature of the heavily underdoped regime is crucial, as this is the region where superconducting pairing begins upon doping an antiferromagnetic Mott insulator. Nonetheless, this region is highly sensitive to disorder owing to the weak screening effect stemming from the low carrier density.

A nearly disorder-free $CuO_2$ environment, which offers a solution, is realized in the inner layers of multilayer cuprates. In the multilayer configuration, these inner layers are spatially separated from the dopant layers, which are the primary source of random Coulomb potentials and lattice distortions. The outer planes effectively screen these disorder potentials, leading to a substantial suppression of both structural distortion and electronic inhomogeneity in the inner $CuO_2$ planes. In particular, the systematic nuclear magnetic resonance NMR studies of multilayer cuprates have demonstrated that in multilayer systems with three or more $CuO_2$ layers per unit cell ($n \geq 3$, $n$ is the number of $CuO_2$ layers per unit cell), the inner $CuO_2$ planes (IPs) exhibit significantly narrower NMR linewidths than not only the outer planes (OPs) but also single- and double-layer cuprates ($n = 1, 2$)[11,12]. This sharp spectrum directly indicates a substantially reduced degree of both lattice and electronic disorder in the IPs. Notably, the highest superconducting critical temperature ($T_c$) among all known materials at ambient pressure has been observed in triple-layer cuprates, where such clean inner $CuO_2$ layers are present[11,13]. Investigating the inner layers of multilayer cuprates is thus likely the key to unlocking a major breakthrough in understanding high-$T_c$ superconductivity; nevertheless, such studies have long been hindered by the difficulty of growing high-quality single crystals.

The argument of "a large Fermi surface vs. a small Fermi pocket" appeared to be resolved by the recent observation of small Fermi pockets in the high-quality single crystals of multilayer cuprates containing five or six $CuO_2$ layers per unit cell[14,15]. However, the superconducting gap for these pockets was very small (~5 meV). Notably, antiferromagnetic (AF) order coexists with superconductivity in multilayer cuprates with $n \geq 3$[11,16]. While AF order may assist in stabilizing Fermi pocket formation, it may also compete with superconductivity, thereby reducing the superconducting gap[17]. Thus, the small superconducting gap observed in the pocket band can be expected. Yet, there remains a possibility that the gap magnitude was small because the doping level is close to the edge of $T_c$-dome[15]. Nonetheless, tuning

doping levels in single crystals of multi-layered systems is highly challenging, and in fact, there has been no such report for the five- and six-layer compounds.

In this article, we focus on the four-layer $Ba_2Ca_3Cu_4O_8(F,O)_2$ (F0234), consisting of only one set of double inner planes per unit cell. This compound allows for a higher doping level than its five- or six-layer counterparts, owing to fewer inner layers being doped by dopant layers. Here we demonstrate that small Fermi pockets can host an substantially large superconducting gap, $\Delta_{pocket} \sim 30$ meV at the pocket tip and $\Delta_0 \sim 60$ meV extrapolated to the antinode. This gap ranks among the largest reported in cuprates and is comparable to that of $HgBa_2Ca_2Cu_3O_{8+\delta}$[18] with the highest $T_c$ in cuprate ($T_c = 130$ K). The strong pairing strength $\Delta_{pocket}/\varepsilon_F \sim 0.6$, along with simultaneously observed pairing pseudogap and Bogoliubov flat-band, all place the inner-layer electronic state in the strong-coupling BCS-BEC crossover regime. Strikingly, the BCS-to-BEC crossover emerges not upon reducing, but rather upon increasing the carrier density, contrary to conventional expectations, and it occurs abruptly within a very narrow doping window. These findings offer the experimental insight into the mechanism of electron pairing that emerges upon doping an antiferromagnetic Mott insulator in a nearly disorder-free $CuO_2$ plane of cuprates, thereby shedding light on the fundamental nature of high-$T_c$ superconductivity.

## Results

A small Fermi pocket has not been observed in four-layer cuprates to date. Here, we present its observation using F0234 with different doping levels of $T_c = 71$ K, 74 K, and 78 K (UD71K, UD74K, and UD78K, respectively; see Fig. 1b). Figure 1c–e show the Fermi surface mapping for the three samples measured by ARPES. The spectral intensities are taken slightly below the Fermi level to avoid spectral suppression due to the superconducting gap. We found a small Fermi pocket, along with a typical Fermi arc characteristic of the underdoped cuprates, in all three samples. In multilayered cuprates, each layer forms a Fermi surface independently of other layers. Therefore, the small pocket and the Fermi arc are each assigned to IPs and OPs, as depicted in Fig. 1a. The Fermi pocket is also confirmed by the de Haas-van Alphen oscillations (Fig. 1g–i; see Supplementary NOTE 2 for more details). Importantly, the oscillation frequencies are nearly identical around 380 T for all the samples, indicating that the doping levels $p$ (the proportion of the pocket size to the whole Brillouin zone) in the IP are nearly the same ($p \sim 5.5\%$) within measurement error of ±0.3%. This is also confirmed in Fig. 1f, which plots the Fermi momentum ($k_F$) points extracted from the ARPES spectra. Both the pockets and arcs for the three samples overlap with each other without a sizable difference. In particular, we find that the Fermi pocket of $p = 5.5\%$ (a black oval) matches well with the ARPES data.

The slight doping differences among the three samples are elucidated by variations in their effective masses. Previous reports have shown that the effective mass increases with doping in the underdoped regime, regardless of their different underlying mechanisms between charge order and Mott physics[14,15,19]. We therefore posit a similar doping-dependent behavior in the four-layer compounds as well. The effective mass ($m^*$) is obtained from the temperature dependence of the quantum oscillation amplitude using the Lifshitz-Kosevich formula (insets of Fig. 1g–i). As depicted in Fig. 1j, we find an increasing trend of $m^*$ with increasing $T_c$. We note that the variation of the effective mass observed in quantum oscillation measurements is subtle, ranging from $m^* \sim 0.82m_0$ in UD71K to ~$0.91m_0$ in UD78K (Fig. 1g–i). In addition, since both the inner and outer $CuO_2$ layers of all three samples remain in the underdoped regime (Supplementary NOTE 5), a higher $T_c$ implies a higher hole concentration. Taken together, these observations indicate the slight increase in carrier density $p$ from UD71K to UD78K. We emphasize, however, that the doping levels of the three samples vary within a very narrow range

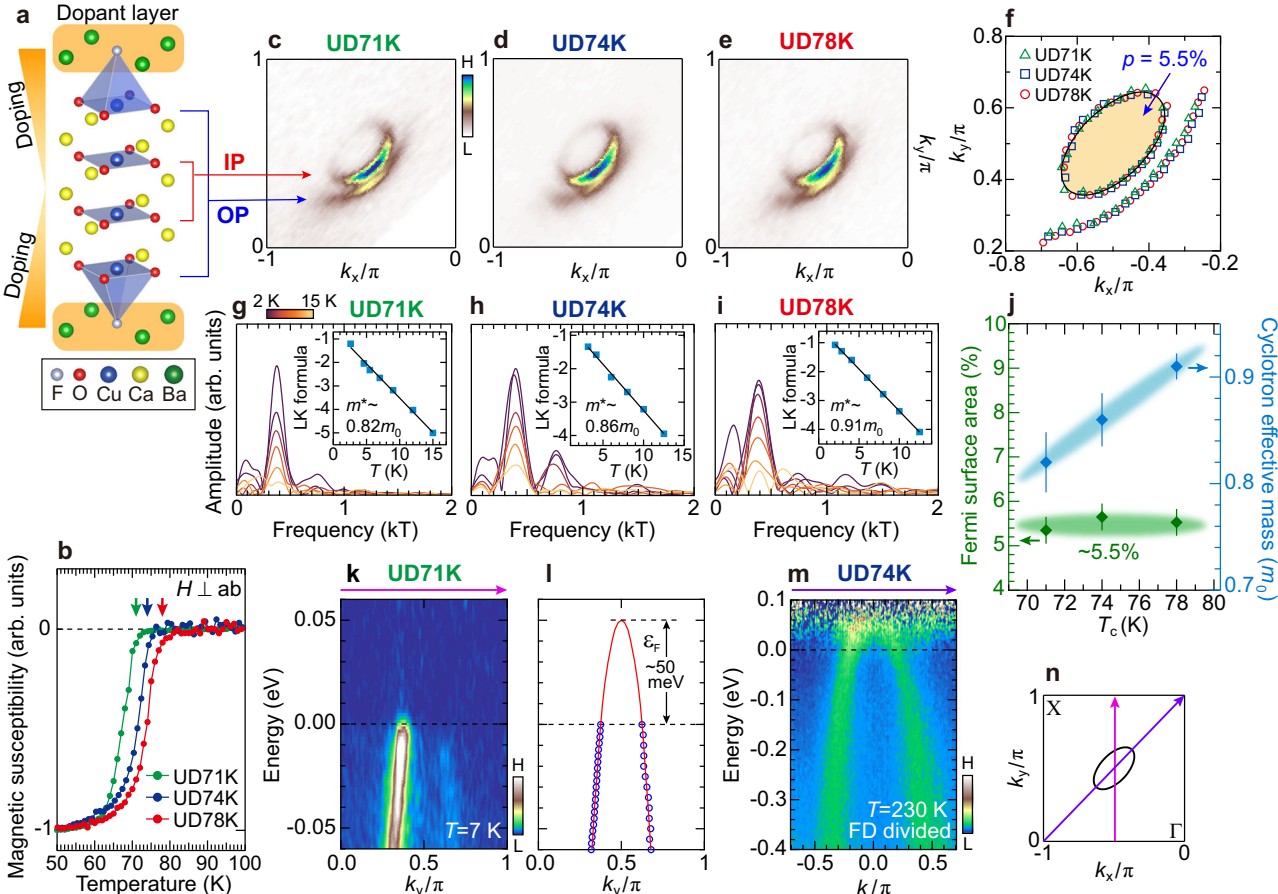

**Fig. 1 | A small Fermi pocket observed by ARPES and quantum oscillation measurements in four-layer cuprates. a** Crystal structure of $Ba_2Ca_3Cu_4O_8(F,O)_2$. **b** Magnetic susceptibility of single crystals (UD71K, UD74K, and UD78K) with $T_c$ values determined from the onset of the Meissner effect (colored arrows). **c**–**e** ARPES Fermi surface (FS) mappings at $T = 7$ K for the three samples. The intensity map at −20 meV, integrated over ±10 meV, avoids the superconducting gap and reveals the FS shape. **f** FSs determined by ARPES. $k_F$ points were determined by identifying the momentum at which the superconducting gap reaches its minimum in EDCs. The black solid line corresponds to a small pocket with area $p = 5.5\%$ estimated from quantum oscillation measurements. **g**–**i** Fast Fourier transformation (FFT) of quantum oscillation spectra for UD71K, UD74K, and

UD78K, respectively, measured from $T \sim 2$ K to ~15 K. Insets show the temperature dependence of FFT peak intensities with fits to the Lifshitz-Kosevich (LK) formula, yielding effective mass ($m^*$). **j** Small pocket area (green diamonds) and effective mass (blue diamonds) both obtained from quantum oscillation data are plotted against $T_c$. **k** ARPES band dispersion of the pocket in UD71K, measured at $T = 7$ K along the magenta arrow in (**n**). **l** Band dispersion determined from the peak positions of momentum distribution curves (MDCs) in (**k**). Red curve represents a tight-binding fit to the data (see Supplementary NOTE 4). **m** ARPES band dispersion in UD74K along the purple arrow in (**n**), measured at $T = 230$ K and divided by the resolution-convoluted Fermi-Dirac function. **n** Momentum cuts for (**k**, **m**) (magenta and purple arrows, respectively) across the Fermi pocket (black line).

(approximately 0.6%) around 5.5%, as estimated from the measurement error (±0.3%) of the quantum oscillation frequency.

As demonstrated in Supplementary NOTE 14 and Fig. S13, the expected change in band dispersion slope associated with the mass variation among three samples is substantially smaller than the intrinsic width of the ARPES spectra, making it impractical to resolve this difference directly by ARPES. Nevertheless, the band dispersion derived from the effective masses obtained from quantum oscillations is in excellent agreement with that observed by ARPES, highlighting the consistency between the two techniques.

A small Fermi pocket corresponds to small $\varepsilon_F$, which is essential for achieving a strong pairing strength ($\Delta_{pocket}/\varepsilon_F$). To accurately characterize the band structure of IP, we perform band-selective measurements, where contamination of the ARPES signal from the outer $CuO_2$ plane (OP) is suppressed by exploiting the matrix element effect in photoemission (see Fig. 4a, b for mappings). Figure 1k, l show the ARPES band map and extracted band dispersion of UD71K at the superconducting state. The pocket band becomes more prominent away from the nodal cut (or diagonal cut), so we select the vertical cut (along magenta arrow in Fig. 1n) from ($\pi/2$, 0) to ($\pi/2$, $\pi$), where the superconducting gap is small enough to minimize its effect on the

band determination. By fitting the occupied band determined by ARPES, we estimate the band top energy ($\varepsilon_F$) as ~50 meV (see Supplementary NOTE 3 for UD74K and UD78K). We also confirm the consistent value of $\varepsilon_F$ via the measurements of a band along the AF zone boundary (AFZB, purple arrow) for UD74K (Fig. 1m). The data of Fig. 1m were taken at a high temperature (230 K) above $T_c$ and divided by the resolution-broadened Fermi-Dirac (FD) function to trace the band dispersion up to $\varepsilon_F$. We note that a sharp parabolic dispersion with well-defined spectral features, as observed along the AFZB here, is never seen for a band with a Fermi arc. Importantly, $\varepsilon_F \sim 50$ meV persists across all three samples.

Next, we measure the superconducting gap along the small pocket. Figure 2a–f present the ARPES spectra for the three samples and their curvature plots[20], which enhance the visibility of peak positions. The spectra were measured along the momentum cut crossing the pocket tip (Fig. 2g, also see Supplementary NOTE 12), where the largest energy gap ($\Delta_{pocket}$) opens in the small pocket. We found that $\Delta_{pocket}$ for UD71K, UD74K, and UD78K are around 15 meV, 19 meV, and 30 meV, respectively. Considering their minimal carrier differences, it is remarkable that the gap size nearly doubles from UD71K to UD78K. Furthermore, the magnitude of this gap is significantly large, albeit

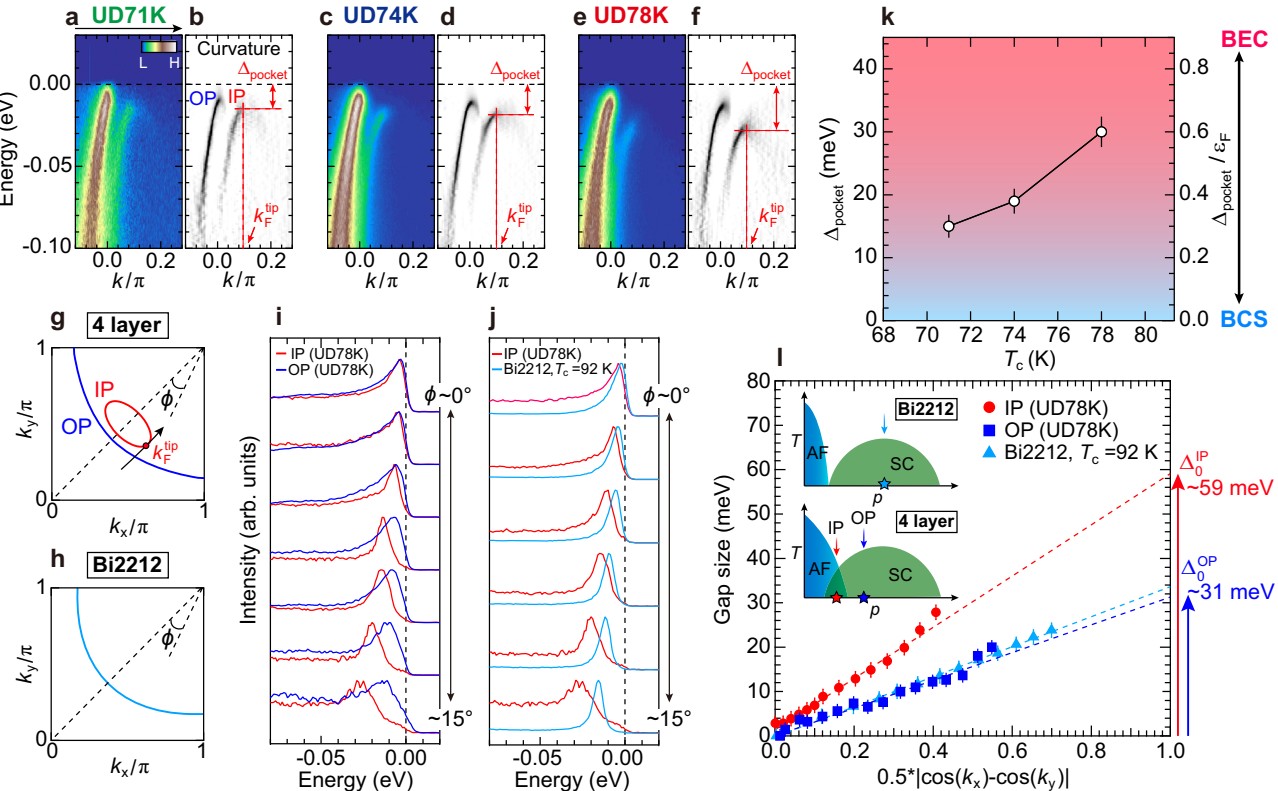

**Fig. 2 | Superconducting gap and pairing strength of the pocket bands. a** ARPES band dispersion of UD71K measured at $T = 10$ K along the black arrow in (**g**), which cross the pocket tip ($k_F^{tip}$). **b** Curvature plot [20] of (**a**): curvature of spectra to clarify peak positions. **c–f** Similar data to (**a**, **b**), but for UD74K and UD78K. $\Delta_{pocket}$ is determined by EDC peaks at $k_F^{tip}$. **g** Schematic FSs for IP (red) and OP (blue) in the four-layer cuprate. **h** Schematic FS of optimally doped Bi2212. **i** EDCs at $k_F$ from $\phi = 0°$ (nodal point) to 15° for IP (red) and OP (blue) in UD78K. **j** Similar data to (**i**), but for IP in UD78K (red) and optimally doped Bi2212 ($T_c = 92$ K) (light blue) [21]. **k** Superconducting(SC) gap at $k_F^{tip}$, $\Delta_{pocket}$, and the corresponding pairing strength

$\Delta_{pocket}/\varepsilon_F$ plotted against $T_c$. The error bars correspond to the estimated variances, considering measurement uncertainties of both the ARPES and quantum oscillation (see Supplementary Information for details). **l** SC gap on each plane in UD78K and Bi2212 plotted against the $d$-wave function $|\cos(k_x) - \cos(k_y)|/2$. Gap values were determined from the symmetrized EDC peaks (see Supplementary Note 10 for a detailed analysis). $\Delta_0$ of IP and OP is extrapolated to the antinode. The inset shows schematic phase diagrams of Bi2212 and IP for the four-layer compound with doping levels indicated by stars and arrows.

with the low carrier concentration of $p \sim 5.5\%$. In Fig. 2j, we compare the energy distribution curves (EDCs) between the IP of the current sample and the optimally doped double-layered $Bi_2Sr_2CaCu_2O_{8+\delta}$(Bi2212) with a higher $T_c$ of 92 K [21], which is the most well-studied cuprate by ARPES. We find that the gap magnitude is about twice as large in IP. In Fig. 2l, we plot the gap size as a function of $d$-wave form $|\cos(k_x) - \cos(k_y)|/2$ (Supplementary NOTE 13). The SC gap extrapolated to the antinode ($\Delta_0$) reaches ~60 meV, which is one of the largest reported in ARPES studies of cuprates [22–29] and is even comparable to that of $HgBa_2Ca_2Cu_3O_{8+\delta}$ [18] with the highest $T_c$ in cuprate ($T_c = 130$ K).

We further find that the gap magnitude of IP is about twice that of OP, as demonstrated in Fig. 2i by overlaying EDCs for IP and OP. Although the OP, being adjacent to the dopant layers, has a higher carrier density than the IP, it remains within the underdoped regime (Supplementary NOTES 5 and 6). Via the investigation of single- and double-layer cuprates [23,24,30], the superconducting gap in the under-doped regime is almost constant or decreases with lower carrier density. Hence, the significant difference in gap magnitude between IP and OP cannot be understood by their doping level difference. We also point out that NMR measurements identified the presence of AF order in the IP for the current compound with similar doping levels to our samples, while it is negligible in OP [12]. The AF order, therefore, does not compete with superconductivity, but rather intimately coexists. Interestingly, the emergence of a large superconducting gap in a small pocket band indicates that a high-$T_c$ superconductivity does not rely on the contributions from the antinodal electronic

states, where the gap magnitude and superfluid density partition are maximal.

A large superconducting gap ($\Delta$) in a small pocket with a small Fermi energy naturally highlights the importance of $\Delta/\varepsilon_F$ [31,32], a dimensionless parameter that defines the pairing strength in super-conductors. Figure 2k plots the pairing strength $\Delta_{pocket}/\varepsilon_F$, calculated using the experimentally observed superconducting gap at the pocket tip ($\Delta_{pocket}$) and $\varepsilon_F$ for the three samples. We obtain remarkably large values ranging from 0.3 to 0.6, which are several orders of magnitude greater than those of conventional BCS-type superconductors, and even exceed those of any cuprate compounds [33]. The pairing strength in the IP of our four-layer cuprates is notably comparable to, or even exceeds, that of bulk and monolayer FeSe [34–37], $Li_xZrNCl$ [38], and magic-angle bilayer graphene [39,40], which are all reported to reside in the BCS-BEC crossover regime (Supplementary NOTE 9).

The core concept of BCS-BEC crossover lies in the formation of tightly bound molecule-like bosonic pairs, which exhibit distinct characteristics compared to classical BCS superconductors. One of the main features is a pseudogap that opens above $T_c$, indicating the pair formation at a higher temperature than that for pair condensation. Here, we note that the pairing pseudogap arises as a precursor to superconductivity [31,32,41], and it should be distinguished from the pseudogap competing with superconductivity that develops near ($\pi$, 0) in a wide doping range of cuprates [22,42,43].

Figure 3a–c show the temperature evolution of EDCs at $k_F$ for the pocket tip ($k_F^{tip}$, see Fig. 2g). The EDCs are symmetrized about the

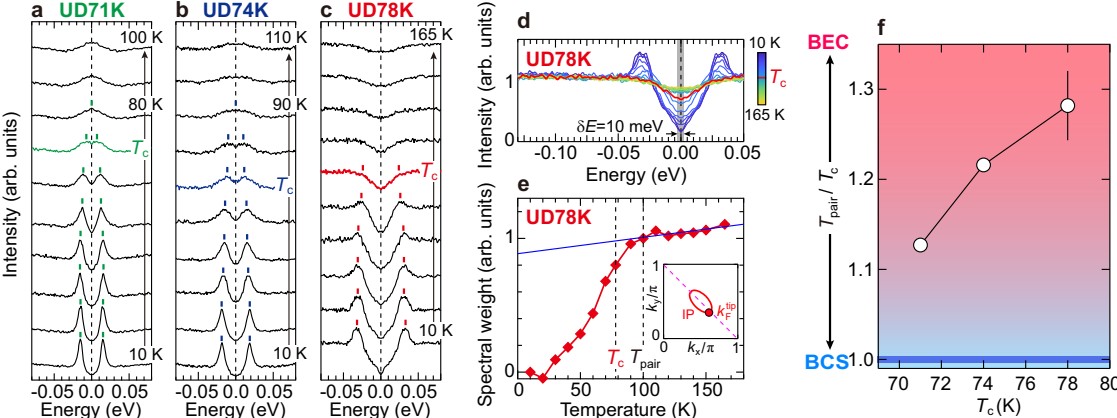

**Fig. 3 | Pairing pseudogap in the pocket and its evolution with carrier doping.** **a**−**c** Temperature variation of symmetrized EDCs at the pocket tip for UD71K, UD74K, and UD78K, respectively. Spectra at $T_c$ are colored. **d** Same data as in (**c**), but plotted without offset. **e** Temperature dependence of spectral weight near the Fermi level, integrated over $\delta E = 10$ meV (gray region in (**d**)). The onset temperature of pairing, $T_{pair}$, is defined as the temperature where the high-$T$ linear behavior at high temperatures is deviated upon cooling. **f** $T_{pair}/T_c$ plotted against $T_c$. The blue line at $T_{pair}/T_c = 1$ indicates the BCS limit, where pair formation and condensation occur at the same temperature. The error bar for UD78K corresponds to the uncertainty in the spectral weight analysis (Supplementary NOTE 15).

Fermi level to eliminate the Fermi cut-off, allowing for the visualization of gap opening or closing. The spectra at low temperatures well below $T_c$ exhibit coherent peaks and the superconducting gap. Intriguingly, a spectral gap persists even above $T_c$ in all samples. For UD71K and UD74K (Fig. 3a, b, respectively), well-defined peaks are observed above $T_c$. This allows us to determine the closing temperature ($T_{pair}$) of the pairing pseudogap: $T_{pair} \sim 80$ K and $T_{pair} \sim 90$ K for UD71K and UD74K, respectively. In contrast, the superconducting peak for UD78K (Fig. 3c) becomes ill-defined with increasing temperature and the gapped state with spectral depletion around the Fermi level continues up to temperatures above $T \sim 165$ K. This indicates that the pseudogap which competes with superconductivity[22,42,43] develops in UD78K due to its slightly higher doping level compared to the other two samples.

To precisely determine $T_{pair}$ for UD78K, we analyze the spectral weight filling within the gap for symmetrized EDCs (Fig. 3d). In Fig. 3e, we plot the spectral weight around the Fermi level (gray region in Fig. 3d) as a function of the temperature. We find that, around 100 K, the rate of increase of the spectral weight changes to a $T$-linear behavior, which is characteristic of the competing pseudogap[44]. From the onset temperature of the deviation, we determine $T_{pair}$ to be ~100 K, which is significantly higher than $T_c$. Figure 3f plots the ratio of $T_{pair}$ to $T_c$ for three samples, showing that the temperature for the preformed pair is raised with an increase of the doping level.

Another notable feature of the BCS-BEC crossover is a Bogoliubov flat band, which signifies the presence of spatially localized pair states. The pair size $\xi$ (or coherence length) is roughly given by $\xi \sim 1/\delta k_F \sim \hbar v_F/2\Delta$ with the momentum span of the flat portions of Bogoliubov bands ($\delta k_F$) expressed as the ratio between the SC gap and the Fermi group velocity ($v_F$). On the other hand, the average inter-pair distance $d_p$ is of the order of $1/k_F \sim \hbar v_F/2\varepsilon_F$ with the Fermi wave number ($k_F$). Therefore, the pairing strength is expressed as their ratio: $d_p/\xi \sim \delta k_F/k_F \sim \Delta/\varepsilon_F$. The flat band that spans from $+k_F$ to $-k_F$ in the pocket indicates a large $\Delta_{pocket}/\varepsilon_F$ of around 1 (that is, the pair size and the distance of pairs becomes comparable), corresponding to the BCS-BEC crossover regime.

The Bogoliubov flat band has been observed in FeSeTe[37] and FeSeS[45] systems, providing evidence of the BCS-BEC crossover. Here, we demonstrate that four-layer cuprates also have such a flat band. We employ the matrix element effect in ARPES to suppress the spectral signals of the OP band, enabling a detailed examination exclusively of the IP band structure. Figure 4a shows the Fermi surface mapping with light polarization aligned along the sample $a$-axis. We found an adequate momentum region where only the pocket band is observed in

the 2nd Brillouin zone. In Fig. 4c–e, we extract the ARPES band dispersions along the momentum Cut 1 (white line in Fig. 4a), which crosses the pocket center horizontally for the three samples. Overall band dispersions are similar in all samples, reflecting their small differences in doping levels.

In Fig. 4h, we present EDCs extracted from the ARPES map for UD78K (Fig. 4e) with an offset for clarity. We find a Bogoliubov flat band spanning from $+k_F$ to $-k_F$, which is consistent with a strong pairing strength $\Delta_{pocket}/\varepsilon_F$. A clear difference among three samples is seen around the pocket center in the prominence of the spectral peak at the pocket center $(\pi/2, \pi/2)$, as shown in Fig. 4i. A coherent peak appears in UD78K (red curve in Fig. 4i), whereas it is less evident in the other two samples (UD71K and UD74K). Overall, the Bogoliubov flat band is most pronounced in UD78K. Notably, the Bogoliubov quasiparticle peak is not expected at $(\pi/2, \pi/2)$, which lies along the nodal direction. A likely cause for a sharp peak is strong coupling with a collective mode, as evidenced by a peak-dip-hump structure in our data (red curve in Fig. 4i, k). The mode energy, estimated to be ~30–40 meV from the peak-to-dip energy spacing, corresponds to a typically observed mode in cuprates[28,46–48]. Our results thus indicate that mode coupling, which is enhanced in the superconducting state, facilitates the generation of a continuous Bogoliubov flat band without a disconnection across the nodal $(\pi/2, \pi/2)$ point. Furthermore, the flat band observed is apparently flatter than a conventional BCS-type Bogoliubov band. This implies that mode coupling renormalizes the band and reinforces the flatness (Supplementary NOTE 11), thereby enhancing the spatial localization of paired electrons.

To confirm that this conclusion is not sensitive to the matrix element effect, we measured the ARPES data for samples rotated by 45 degrees in the azimuth angle, maintaining light polarization with respect to the analyzer. The obtained Fermi surface map is displayed in Fig. 4b. We find another adequate momentum region in a different 3rd Brillouin zone, where only the pocket signals are pronounced. For two samples, UD74K and UD78K, the band dispersions are extracted along the diagonal direction of Cut 2, which crosses the pocket center diagonally (or along AFZB), as represented in Fig. 4b with a white line. Again, we find a coherent peak at the pocket center in UD78K, which is instead unclear in UD74K (Fig. 4k). A Bogoliubov flat band is also reproduced in Fig. 4j, which plots EDCs for the ARPES map of Fig. 4g. These consistent results, which are robust against the matrix element effect, clearly demonstrate that the flat band is an intrinsic property and that UD78K is particularly closer to the BEC regime.

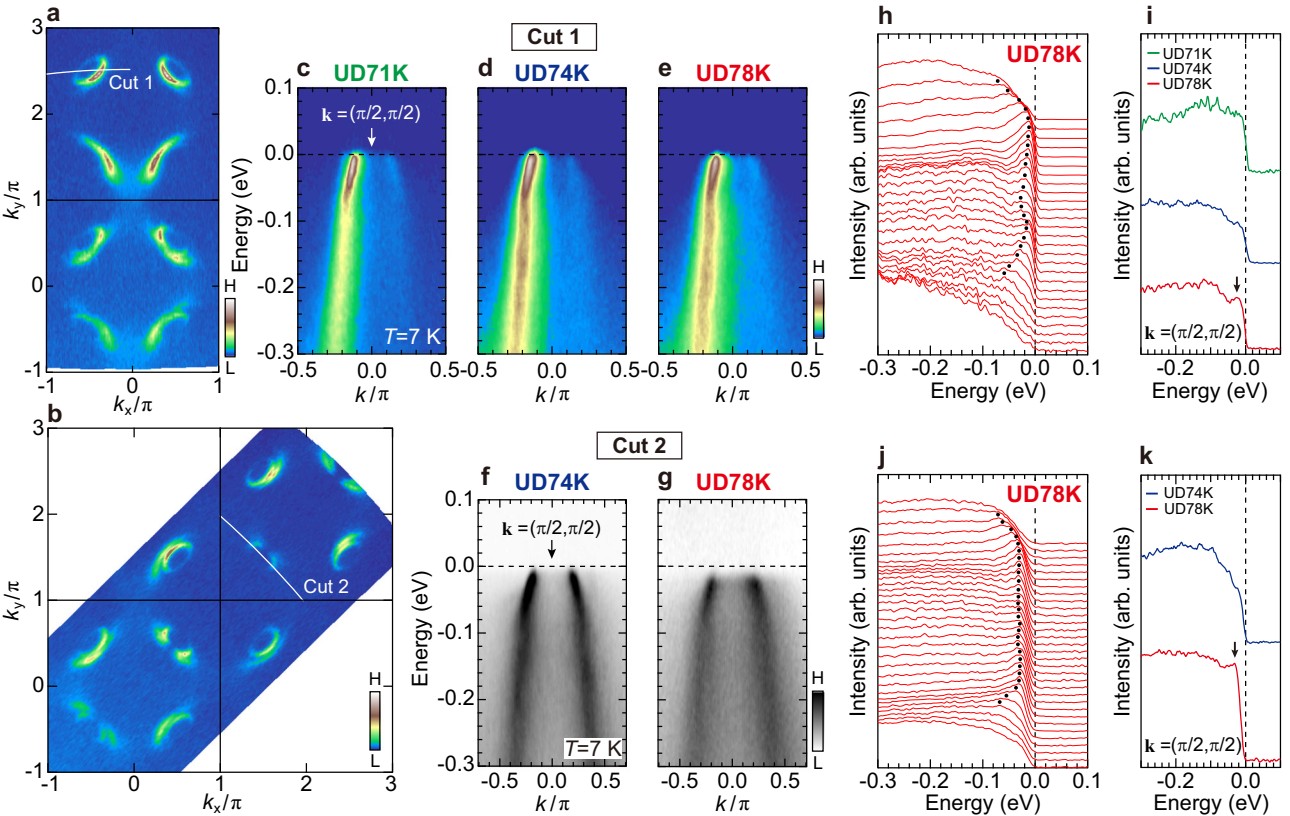

**Fig. 4 | Bogoliubov flat band in the pocket. a, b** ARPES FS maps of UD71K measured with different experimental geometries. **c–e** ARPES band dispersion for UD71K, UD74K, and UD78K, measured along Cut 1 in (**a**). **f, g** Similar data to (**c–e**), but measured along Cut 2 in (**b**). **h** Full set of EDCs in (**e**). Peak positions (black circles) trace a Bogoliubov flat band. **i**, EDCs at the pocket center ($\vec{k} = (\pi/2, \pi/2)$) extracted from (**c–e**). **j** Same as (**h**), but for (**g**). These results consistently indicate a Bogoliubov flat band in the pocket. **k** Same as (**i**), but for (**f, g**).

Our findings are summarized in Fig. 5. We found that a small pocket and a large superconducting gap simultaneously emerge in cuprates. The AF order coexists with superconductivity in the inner plane that hosts a small pocket, yet its superconducting gap is significantly larger than that of the paramagnetic outer plane with a Fermi arc by a factor of two (Fig. 4l and Fig. 5a–c). This indicates that the AF order does not compete with the high-$T_c$ conductivity but rather has an intimate relationship.

In Fig. 5d–e, we illustrate an electronic phase diagram of multilayer cuprates based on our results. Although the doping variation we addressed is merely within 0.6 %, a drastic variation occurs. Whereas the two temperatures of $T_{pair}$ and $T_c$ are going to merge towards the edge of the superconducting dome, the $T_{pair}$ is much more rapidly increased at higher doping levels than the $T_c$. This causes a clear discrepancy between the electron pairing and the superconducting condensation, highlighting a rapid approach toward the BEC regime in the inner layer of the four-layered cuprates. The well-acknowledged superconducting phase diagram in cuprates suggests that the BCS to BEC crossover occurs with decreasing carrier concentration from the overdoping to underdoping regimes[33,49,50]. Interestingly, our data suggest a quite different trend: the crossover from BCS to BEC occurs with increasing doping level, contrary to the general expectation.

Finally, we categorize the superconductivity of the four-layer cuprate in the Uemura plot[33], which plots $T_c/T_F$ ratio of superconductors, where $T_F$ is the Fermi temperature. For such categorization, it is crucial to confirm which layer dominantly determines the $T_c$ value in our four-layer system. We obtained the following spectroscopic signatures indicating that the pocket band in IPs determines the $T_c$ in the system. The extrapolated gap $\Delta_0$ in the IPs is approximately twice that in the OPs (Fig. 2l and Supplementary NOTE 6). Consistently, quasiparticle coherence peaks are markedly sharper and more pronounced for the IPs, implying reduced scattering and a more robust pair condensation (Supplementary NOTE 7). Most importantly, the OP gap closes at a temperature below the $T_c$, whereas the IP gap persists up to temperatures higher than $T_c$ (Supplementary NOTE 8)[51,52]. In light of these observations, we posit that the onset of superconductivity in our samples is determined by the inner layers. According to the Uemura plot[33], most unconventional superconductors, including typical cuprates, heavy fermions, and iron chalcogenides, exhibit $T_c/T_F \sim 0.05$, which is considerably distant from conventional BCS-type superconductors (Fig. 5f). Remarkably, the clean inner planes of the current four-layer cuprates show a much larger ratio ($T_c/T_F \sim 0.13$ for UD78K) than other unconventional superconductors, nearly reaching the theoretical upper bound expected in two-dimensional (2D) superconductivity ($T_c/T_F = 0.125$)[53].

## Discussion and outlook

The coexistence of antiferromagnetic (AF) order and superconductivity is an intrinsic property of disorder-free underdoped cuprates, as consistently observed in the inner layers of multilayer systems with three or more $CuO_2$ planes per unit cell[11,14–16]. In such nearly ideal $CuO_2$ planes, superconductivity develops within a doped AF-Mott state while coexisting with static AF order, a regime that cannot be realized in single- or double-layer cuprates where disorder is unavoidable.

The crossover from BCS- to BEC-like behavior with increasing carrier density may appear counterintuitive. A natural explanation is that static AF order suppresses low-energy magnetic fluctuations,

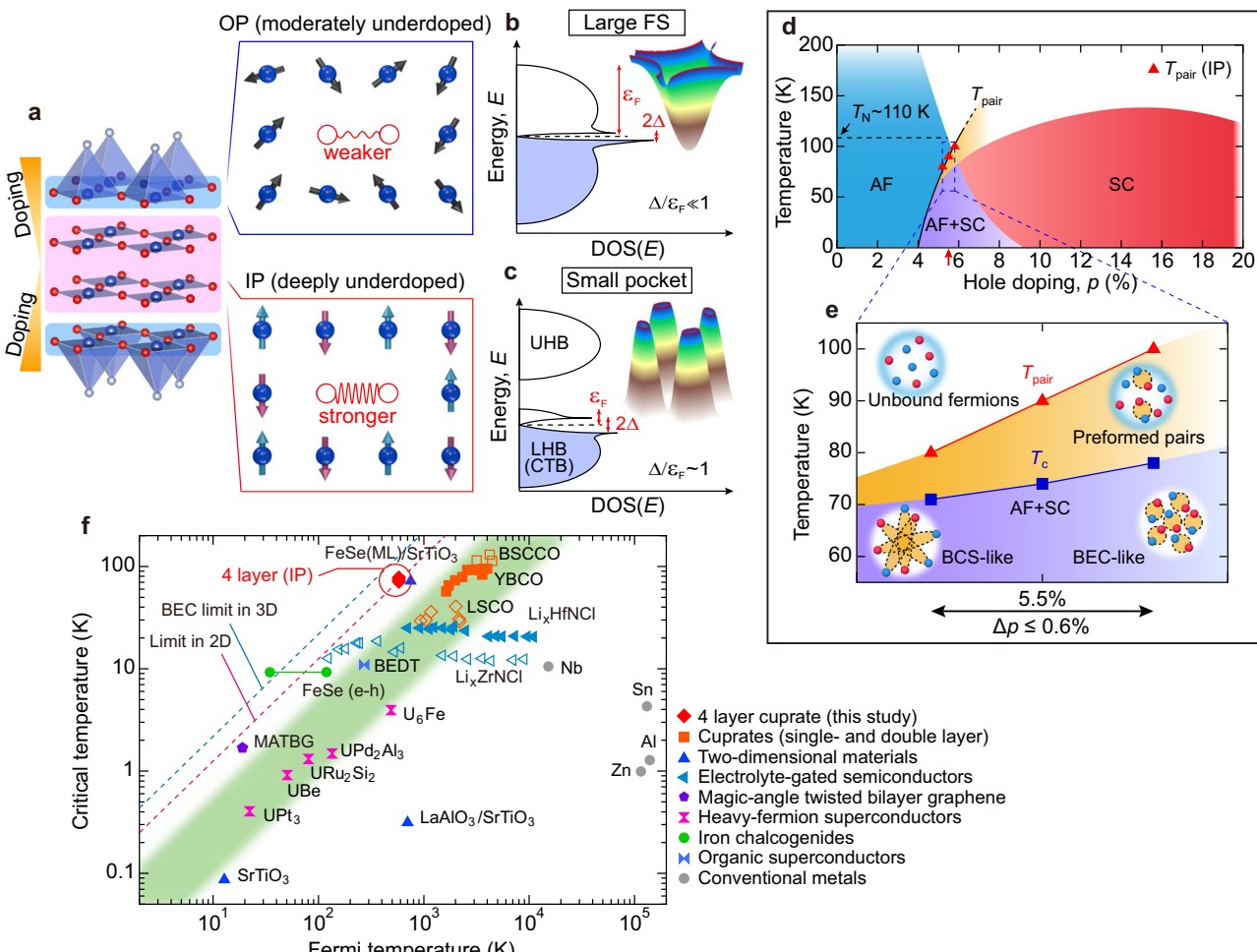

**Fig. 5 | Strongly bound pairs and phase diagram of the inner CuO₂ planes.**
**a** Schematic illustration of pairing strengths in each $CuO_2$ plane of four-layer cuprates. The IP forms a small Fermi pocket, is less doped and farther from optimal doping than the OP. The IP develops AF order, unlike the optimally-doped, paramagnetic OP with a large FS (or Fermi arc). Despite these features being seemingly unfavorable for stabilizing pairing, the SC gap of the IP is significantly larger than that of the OP. **b** Schematic density of states (DOS) and band structure in the OP, featuring a large FS with a large Fermi energy $\varepsilon_F$ (~1 eV) and a small SC gap (~30 meV)[3,61], resulting in a weaker pairing strength, $\Delta/\varepsilon_F$ ~ 0.03 (≪1). **c** Schematic DOS and band structure in the IP: carriers are doped into the lower Hubbard band (LHB) or charge transfer band (CTB), generating a small Fermi pocket with a small Fermi energy $\varepsilon_F$ (~50 meV) and a large SC gap ($\Delta_{pocket}$ ~ 30 meV at the pocket tip and $\Delta_0$ ~ 60 meV at the antinode). The SC gap and small $\varepsilon_F$ become comparable, yielding a stronger pairing strength $\Delta_{pocket}/\varepsilon_F$ ~ 0.6, which is much larger than $\Delta/\varepsilon_F$ ~ 0.03 in

OP. **d** Phase diagram of the inner $CuO_2$ planes. AF phase and superconducting dome are referenced from NMR studies on the same compound[16]. The purple region denotes coexisting AF order and superconductivity. The red arrow marks the doping level of our samples ($p$ ~ 5.5%). Red triangles represent the $T_{pair}$ values, evolving alongside the superconducting dome. The phase of preformed pairs is shaded in orange. **e** Enlarged view of (**d**) (blue dashed box) focusing on the region around $p$ ~ 5.5%, summarizing our results and illustrating the BCS-BEC crossover from the BCS side (UD71K) to the BEC side (UD78K). **f** Uemura plot: A logarithmic plot of critical temperature $T_c$ versus Fermi temperature $T_F$ for various superconductors[33,36,38–40,62–64]. The green shaded area indicates the region where most unconventional superconductors lie. The IPs of the four-layer cuprates, which form a small Fermi pocket, are marked by red diamonds; they reach the upper bound of the two-dimensional (2D) Berezinskii-Kosterlitz-Thouless (BKT) transition ($T_c/T_F = 0.125$)[53].

thereby stabilizing a more BCS-like regime even in a small Fermi pocket with low carrier density. As doping weakens the AF order, the magnetic exchange energy becomes comparable to the electronic kinetic energy associated with the pocket. Because the pocket has an intrinsically low $\varepsilon_F$, even a slight increase in carrier density (only ~0.6%) can substantially modify the balance between static order and dynamical fluctuations, enhancing the pairing scale relative to $\varepsilon_F$ and pushing the system toward the BEC side of the crossover.

The inner $CuO_2$ planes thus lie in a regime where magnetic and kinetic energy scales are finely balanced. Such sensitivity suggests proximity to an antiferromagnetic quantum critical regime[54], in which minute doping-induced changes can qualitatively alter the superconducting state. Our results demonstrate that in nearly ideal $CuO_2$ planes, tiny carrier tuning can switch superconductivity between BCS- and BEC-like regimes while preserving a well-defined small Fermi

pocket. While other interactions, including electron-phonon coupling, may also contribute, magnetic fluctuations are expected to be strongly enhanced near AF instability and can influence the pairing scale.

The band structure in the antinodal region is characterized by low group velocity and therefore contributes significantly to the density of states and to the superfluid density in the superconducting state of cuprates. However, the pocket band lacks the contribution of antinodal states to pairing. Our results, therefore, demonstrate that this seemingly advantageous feature of the electronic structure in high-$T_c$ superconductivity is not essential for realizing a large superconducting gap and strong pairing strength. One possible scenario leading to this situation is that the pseudogap state, which competes with superconductivity[22,42,43], is largely suppressed when a pocket band is formed, as its electronic states disperse down to ~1 eV far from the Fermi level and the low-energy electronic states required for the

development of the pseudogap are absent in the antinodal region[14]. This could make electron pairing more stable in the pocket band state than in the Fermi arc state observed in underdoped single- and double-layer cuprates.

To clarify the unique position of cuprates in the broader context of BCS-BEC crossover research, we briefly compare them with other material systems. Fe-based superconductors have played a pioneering role as the only platform in which the BCS-BEC crossover has been directly identified via direct band observations[37,45]. However, its multi-orbital nature makes the BCS-BEC crossover scenario rather complicated, as the $\Delta/\varepsilon_F$ ratio decreases toward the BEC regime[45]. Layered nitrides[38], which are doped semiconductors, serve as conceptually simple platforms with $s$-wave pairing and minimal orbital complexity; in contrast, the exotic nature due to strong correlation is not induced. Organic superconductors[55] highlight the potential of low-dimensional correlated systems, though their surface sensitivity makes ARPES experiments challenging. Magic-angle twisted bilayer graphene[39,40] provides a fascinating Mott-related system, albeit operating at energy and temperature scales more than an order of magnitude lower than those of cuprates. Within this diverse landscape, cuprates stand out as a uniquely powerful platform, combining a single-band, $d$-wave symmetry, strong Mott-derived correlations, and high-temperature, large-energy scales. Moreover, the emergence of a well-defined pocket band in a clean inner $CuO_2$ layer enables direct comparison with theoretical models, offering an ideal setting for exploring the microscopic nature of the BCS-BEC crossover in strongly correlated systems. It is worth noting that the clean $CuO_2$ layer where the superconductivity coexists with AF order is likewise present in three-layer cuprates as well[11,16,56], which exhibits the highest $T_c$ among all cuprate families with varying numbers of $CuO_2$ layers per unit cell[57]. This correspondence indicates that our results are not limited to a particular case, but reflect the intrinsic physics of clean $CuO_2$ planes, a key for elucidating the pairing mechanism of cuprate superconductors. Collectively, these features establish the multilayered cuprates as a uniquely positioned benchmark system for future experimental and theoretical investigations into strongly coupled superconductivity.

## Methods
### Samples
Single crystals of underdoped $Ba_2Ca_3Cu_4O_8(F,O)_2$ (crystal structure is provided in Fig. 1a) with $T_c = 71$ K, 74 K, and 78 K were grown at between 1000 °C and 1200 °C under a pressure of 4.5 GPa without an intentional flux. The starting composition for the crystal synthesis is $Ba_2Ca_{2.1}Cu_{3.1}O_{6.3}F_2$, which is known to be almost the same in the single crystals[58]. We have conducted X-ray diffraction measurements along the $c$-axis for all the sample pieces and confirmed that they are single crystals, not the mixtures of crystal domains with different numbers of $CuO_2$ layers per unit cell. Each single crystal piece from the batch exhibited a slight variance in the $T_c$. We characterized every single crystal piece by measuring magnetic susceptibility (Fig. 1b) and classified them into the three doping levels, according to their $T_c$. Laue image of the single crystal (Supplementary NOTE 1) shows a four-fold rotational symmetry with no indication of structural modulations.

### ARPES measurements
Laser-based ARPES data were accumulated using a laboratory-based system consisting of a Scienta R4000 electron analyzer and a 6.994 eV laser (the 6th harmonic of Nd: $YVO_4$ quasi-continuous wave) at the Laser and Synchrotron Research Center at Institute for Solid State Physics (ISSP), the University of Tokyo. The data presented are measured at 10 K. The overall energy resolution in the ARPES experiment was set to 1.6 meV. Synchrotron-based ARPES measurements were performed at the high-resolution branch (HR-ARPES) of the beamline I05 in the Diamond Light Source[59], equipped with an MBS A-1 analyzer,

and ARPES end station at Stanford Synchrotron Radiation Lightsource beamline 5–2, equipped with a Scienta DA30 analyzer. The data presented are measured at the photon energy of 55 eV and at the temperature of 7 K. The overall energy resolution was set to ~10 meV in our experiments. In both the laser- and synchrotron-ARPES measurements, a typical cleavage method was used to get a clean surface of the samples: a top post glued on the crystal is hit in situ to obtain a flat surface suitable for the ARPES measurements. The cleavage plane has been confirmed by scanning tunneling microscopy to be along the F/O dopant layers[58]. The superconducting gap is obtained by fitting the ARPES spectra to the phenomenological model (Supplementary Information).

### Quantum oscillation measurements
Torque magnetometry experiments were performed using a commercial piezoresistive cantilever (SEIKO PRC-120)[60] in pulsed magnetic fields up to 60 T (36 ms pulse duration) at the International Megagauss Laboratory at ISSP, the University of Tokyo. The cantilever directly detects the magnetic torque ($\tau$) as the result of the anisotropic magnetization of the sample, $\tau = \boldsymbol{M} \times \boldsymbol{H}$, and the magnetic quantum oscillation known as the de Haas-van Alphen (dHvA) oscillation was observed. According to the Onsager relation, the quantum oscillation frequency $F$ is directly proportional to the extremal cross-sectional area of the Fermi surface perpendicular to the applied magnetic field, $S$, via $F = \hbar S/2\pi e$. In our measurements, the magnetic field was applied nearly parallel to the crystallographic $c$-axis, so that the oscillation frequency reflects the in-plane Fermi surface area (in the $ab$-plane).

In torque magnetometry measurement, the sign of the torque reverses when the magnetic field angle crosses the crystallographic $c$-axis. Therefore, we first determined the torque reversal angle through several low-field measurements. Based on this procedure, we aligned the sample $c$-axis nearly parallel to the external field direction, making the cross-sectional Fermi pocket area close to the area projected onto the cleave plane ($ab$-plane), which was used in the ARPES measurement. Here, the uncertainty of the field angle relative to the $c$-axis ($\theta$) is estimated to be approximately ±1°. This level of angular uncertainty does not introduce a significant error in the estimation of the Fermi surface cross-sectional area with respect to the external field direction, since the angle-dependent quantum oscillation measurement revealed that the quantum oscillation frequency (cross-sectional area) of the small Fermi pocket follows a ($\cos\theta$) form, namely, two-dimensional-like behavior[14]. Supplementary Fig. 2 shows the magnetic torque signals after subtracting background, which was obtained by fitting a polynomial function to each curve of the raw data in the range of magnetic field between 37 and 60 T.

## Data availability
The data that support the findings of this study are available from the corresponding authors upon request.

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

## Acknowledgements

We thank K. Mehlawat and T. Shitaokoshi for their help with the quantum-oscillation measurements, T. Yajima and J. Yamaura for technical assistance with the X-ray measurements, and T. Yamauchi for help with the magnetic-susceptibility measurements. We thank Diamond Light Source for access to beamline I05 under proposals SI36822, SI30646, SI28930, SI25416, and Stanford Synchrotron Radiation Lightsource for access to beamline BL5-2 under proposal S-XV-ST-6368A that contributed to the results presented here. T.K. discloses support for the research of this work from JSPS KAKENHI [Grant numbers: JP21H04439, JP23K17351, JP25H01250, JP25H01246, and 26H02014], the Asahi Glass Foundation, MEXT Q-LEAP [Grant number: JPMXS0118068681], The Murata Science Foundation, The Mitsubishi Foundation, and Toray Science Foundation. K.T. discloses support for publication of this work from JSPS KAKENHI [Grant number: 24K06965]. T.T. acknowledges the support received for this work from JSPS KAKENHI [Grant numbers: 24K00560 and 25H01248]. C.L., M.H., and D.L. acknowledge the support of the U.S. Department of Energy, Office of Science, Office of Basic Energy Sciences, Division of Material Sciences and Engineering, under Contract No. DE-AC02-76SF00515.

## Author contributions

T.K. conceived and designed the project. J.J and K.K. performed the ARPES experiments with the help from K.K., S.H., M.D.W., T.K.K., C.C., C.L., M.H., D.L. and T.K. J.J analyzed the ARPES data with the help from K.K. and T.K. Y.E., T.N., K.A. and K.T. grew the crystals, and J.J., K.K., Y.E., T.Na., K.A., and K.T. conducted the sample characterization. J.J., K.K., and Y.K. performed the quantum oscillations experiments with help from Z.Y. and T.No. J.J. and K.K. analyzed the quantum oscillation data with the help from Y.K. J.J., K.K., S.S., T.T., K.T. and T.K. interpreted the data. All authors discussed the results, and J.J. and T.K. wrote the manuscript. T.K. and K.T. supervised the overall project.

## Competing interests

The authors declare no competing interests.
