## [Transparent Peer Review file · Nature Communications]

BCS-BEC crossover driven by small Fermi pockets of a high- T_c cuprate superconductor

Corresponding Author: Professor Takeshi Kondo

Version 0:

Reviewer comments:

Reviewer #1

(Remarks to the Author)

In the work of Junhyeok Jeong et al. entitled "Extremely large superconducting gap in a small Fermi pocket of high- T_c cuprates", the authors conduct systematic ARPES measurements on four-layer cuprate samples F0234 which consist of two disorder-free inner CuO₂ planes. With doping-, momentum- and temperature-dependent investigations on the electronic structures, the results reveal a large superconducting gap at the small Fermi pocket and an extraordinary behavior in BCS to BEC crossover. Since the small Fermi pocket is closely related to the disorder-free inner CuO₂ layers, above observations thus could be viewed as the intrinsic properties of CuO₂ planes and give new insight into the pairing mechanism of cuprates in underdoped regime. This work addresses a problem with high scientific value, however, before being published, some technical questions still need to be answered seriously.

Q1. Although the authors claim that the inner CuO₂ planes are disorder-free throughout the paper, no direct evidence (e.g. TEM image) is presented to confirm the lattice quality. The authors should give more arguments to justify the lattice quality of the inner CuO₂ planes.

Q2. Identifying the superconducting gap is critical to this work, however, it is not clear how the gap is obtained in the paper. The authors should clarify the quantitative procedure in detail for extracting the superconducting gap from the EDCs.

Q3. The justification of the Bogoliubov flat band in Fig. 4 is weak, because there is high probability that the flat band is contributed by the disorders since the lattice quality has not been confirmed yet (see Q1). To rule out above possibility, a contrast experiment (below T_{pair} vs. above T_{pair}) is needed.

Q4. The authors come up with a scenario accounting for the BCS-BEC crossover that slight increase in carrier density can sensitively shift the balance between AF order and magnetic fluctuations. But it still lacks a clear picture about how that happens. A qualitative mechanism, at least, should be provided.

Reviewer #2

(Remarks to the Author)

In this work, Junhyeok Jeong and coauthors analyze the electronic structure of four-layer cuprate superconductor Ba₂Ca₃Cu₄O₈(F,O)₂ using ARPES and quantum oscillation measurements. The main experiment findings are the observations of an unusually large gap on the small pocket and the observation of a flat Bogoliubov band. The authors argue that these experimental findings are an indication of the BCS-BEC crossover in this material. In general, the paper demonstrates high-quality data and deep analysis. I think it can be published in "Nature Communications" after the following issues are addressed.

1) I don't think the spectra in Fig. 2e-f were a good choice to demonstrate the gap size on the small pocket. If these spectra are measured slightly off the pocket tip, this band with a large gap may just be a band with the top locked below the FL. The best way to address this would be to compare these spectra with the identical spectra measured at $T > T_c$. So the reader would see that this band is actually crossing the FL in the normal state. Another option is to add the spectra measured in the orthogonal direction. The zoomed-in sections of Fig. 4(f, j) may work.

- 2) The authors present the results of the gap-size analysis only for a quarter of the pocket. It would be nice to see experimental evidence that the gap on the outer part ($|k_x|+|k_y| > \pi$) of the pocket also follows a d-wave form.
- 3) The authors provided reasonable evidence for the presence of the Bogoliubov flat band in the UD78K sample. However, I find the phrase "it is less evident in the other two samples" and the arrows near green and blue EDCs in Fig. 3(i, k) misleading. I see no signs of the Bogoliubov flat band in the UD74K, UD74K data. The authors should either clearly state that there is no evidence for the Bogoliubov flat band in these 2 samples and remove the misleading arrows, or provide less ambiguous data. The comparison of EDCs below and above T_c would be very beneficial.
- 4) The gap size on the small pocket in this material is indeed substantially larger than in $\text{Ba}_2\text{Ca}_4\text{Cu}_5\text{O}_{10}(\text{F},\text{O})_2$ and $\text{Ba}_2\text{Ca}_5\text{Cu}_6\text{O}_{12}(\text{F},\text{O})_2$. However, this 30 meV gap is not uncommon for cuprates. Thus, calling this gap "extremely large" is misleading. The title should be changed.
- 5) The authors should add references to support the following claims:
 "The highest superconducting critical temperature among all known materials at ambient pressure has been observed in triple-layer cuprates, where such clean inner CuO_2 layers are present."
 "While AF order may assist in stabilizing Fermi pocket formation, it may also compete with superconductivity, thereby reducing the superconducting gap."
- 6) The following sentence in the supplementary should be rephrased. "The SC gap in IP persists up to the bulk $T_c=74$ K (pink curves) and closes at $T = 90$ K." It sounds like the gap that closes at 90 K is the superconducting gap.

Reviewer #3

(Remarks to the Author)

Junhyeok et. al. have conducted high resolution angle resolved photoemission spectroscopy (ARPES) measurements on a four-layer copper-based high- T_c superconductor $\text{Ba}_2\text{Ca}_3\text{Cu}_4\text{O}_8(\text{F},\text{O})_2$. According to the report, the doping for the inner plane was very underdoped. This makes this study novel as it was previously challenging to stabilize underdoped copper-based superconductor in other previous cuprate materials such as YBCO.

They have observed several signatures of BCS-BEC crossover with their magnetic quantum oscillation and ARPES measurements. The signatures are 1. Coexistence of large superconducting gaps and small Fermi surface pocket; 2. Ratio between pair forming temperature (T_{pair}) and critical temperature (T_c) larger than 1. 3. Observation of signature for a flat band near Fermi level.

While the study is of potential interest and impact to the research of high- T_c superconductor and potentially merits publication in Nature Communications, I find there are several questions that should be addressed before publication:

1. In Fig. 1f, the Fermi surface pocket is determined in a Fermi surface map. How are the data points determined? Is it in EDC or MDC? How does the area of the inner pocket determined by ARPES relate to the quantum oscillation frequency?
2. As the effective mass is observed to change as the T_c (doping level) changes, is it reflected in the ARPES, for example in the slope of the band?
3. In Fig. 3f, the authors reported T_{pair} / T_c as a function of T_c (doping). Why there is no errorbar to these datapoints? According to the caption, the T_{pair} is defined as "the temperature where the high-T linear behavior at high temperatures is deviated upon cooling", however, in Fig. 3e, there seems only one data point that can tell the deviation from the high-T linear behavior, which should give a large error in T_{pair} / T_c . How do these criteria of determining T_{pair} relate to the way reported in supplementary material Fig. S8?
4. The T_c was determined by the onset of magnetic transition, which is in principle measuring the part of the sample which firstly give superconducting screening. How are you sure T_c determined in this way corresponds to inner plane T_c rather than outer plane T_c , which is reported to have higher hole doping and should have higher T_c ?

I also have two minor / technical questions to raise:

1. In the piezo cantilever measurement, how well is the field orientation determined? What is the error between the quantum oscillation orbit and the cleave plane for ARPES measurement?
2. The x-axis label for the Fig. 5e should be p instead of p

Version 1:

Reviewer comments:

Reviewer #1

(Remarks to the Author)

All my concerns are addressed, and I agree that this paper can be published.

Reviewer #2

(Remarks to the Author)

The authors have reasonably addressed my comments. I believe this manuscript can be published after the following minor issue is addressed.

In response to my comments #1 and #2, the authors added supplementary notes 12 and 13. However, they did not refer to them in the main text. Reference to note 12 should be added to the paragraph where Fig. 2a-f are discussed. And reference to note 13 should be added to the paragraph where the d-wave shape of the IP gap is discussed (the same paragraph).

Reviewer #3

(Remarks to the Author)

The authors have done a very good job of addressing the previously raised concerns. I now recommend publication in Nat Comms

Reviewer #4

(Remarks to the Author)
